Evaluation of the global impacts of mitigation on persistent, bioaccumulative and toxic pollutants in marine fish

Bonito Lindsay T. lbonito@ucsd.edu
Hamdoun Amro
Sandin Stuart A. ssandin@ucsd.edu
Scripps Institution of Oceanography, University of California, San Diego , La Jolla, CA , United States of America
Sanderson J. Thomas
Electronic publication date: 2016 Jan 28
Publication date: 2016
Volume: 4
Electronic Location ID: e1573
Received 2015 Jul 3; Accepted 2015 Dec 16
Copyright: ©2016 Bonito et al.
Copyright year: 2016
Copyright holder: Bonito et al.
License: This is an open access article distributed under the terms of the Creative Commons Attribution License, which permits unrestricted use, distribution, reproduction and adaptation in any medium and for any purpose provided that it is properly attributed. For attribution, the original author(s), title, publication source (PeerJ) and either DOI or URL of the article must be cited.
License URL: https://creativecommons.org/licenses/by/4.0/

Keywords: Contaminants, Marine fish, Muscle tissue, Seafood, Persistent organic pollutants, Global, Toxins

Funding: US National Science Foundation NSF 1314480 This research was supported by a grant from the Waitt Foundation, and additional support was provided by the US National Science Foundation (NSF 1314480). The funders had no role in study design, data collection and analysis, decision to publish, or preparation of the manuscript.

==============================
Although persistent, bioaccumulative and toxic pollutants (PBTs) are well-studied individually, their distribution and variability on a global scale are largely unknown, particularly in marine fish. Using 2,662 measurements collected from peer-reviewed literature spanning 1969–2012, we examined variability of five classes of PBTs, considering effects of geography, habitat, and trophic level on observed concentrations. While we see large-scale spatial patterning in some PBTs (chlordanes, polychlorinated biphenyls), habitat type and trophic level did not contribute to significant patterning, with the exception of mercury. We further examined patterns of change in PBT concentration as a function of sampling year. All PBTs showed significant declines in concentration levels through time, ranging from 15–30% reduction per decade across PBT groups. Despite consistent evidence of reductions, variation in pollutant concentration remains high, indicating ongoing consumer risk of exposure to fish with pollutant levels exceeding EPA screening values. The temporal trends indicate that mitigation programs are effective, but that global levels decline slowly. In order for monitoring efforts to provide more targeted assessments of risk to PBT exposure, these data highlight an urgent need for improved replication and standardization of pollutant monitoring protocols for marine finfish.

Introduction

Persistent, bioaccumulative, and toxic pollutants (PBTs), namely polychlorinated biphenyls (PCBs), polybrominated diphenyl ethers (PBDEs), organochlorine pesticides (DDTs and CHLs), and mercury (Hg, MeHg), were widely used throughout the globe in the past century. Although their propensity to bioaccumulate up the food web is well documented, their distribution within seafood across large spatial scales is largely unknown. Nearly 3 billion people rely on fish for their major source of protein (FAO, 2014), suggesting that a significant proportion of the world’s population is exposed to PBTs through seafood (Asplund et al., 1994; Schade & Heinzow, 1998; Gobeille et al., 2006; Domingo & Bocio, 2007; Schecter et al., 2010a; Schecter et al., 2010b). Despite the many potential health benefits from seafood consumption, a risk of pollutant exposure remains.

Assessing the global fate of PBTs is critical for understanding exposure and human health risk to these chemicals. Challenges to such assessments include the costly nature of toxicological sampling and analysis, as well as the high diversity of sources of PBTs that complicates prediction of accumulation patterns, transportability, and persistence. Estimates of global emissions and secondary sources of PBTs are limited to non-existent for certain groups (prior to 2000), limiting the accuracy and availability of global mass budget estimates (Jones & De Voogt, 1999; Breivik et al., 2002a; Breivik et al., 2002b; Breivik et al., 2007; Lohmann et al., 2007). Furthermore, with insufficient knowledge of biogeochemical cycles and geophysical drivers of pollutant transport, partitioning, and accumulation, a global assessment of human toxin exposure is quite difficult to quantify. To address these limitations, several literature reviews have been conducted to assess the spatial extent of persistent contaminants around the world. Although the quantity of publications is notable, none of these reviews have addressed pollutants screened in food fish on a global scale. Rather, reviews conducted thus far have focused on specific pollutant classes in specific regions, most commonly brominated flame retardants in the Arctic (Law et al., 2003; Kallenborn, 2006; Yogui & Sericano, 2009; De Wit, Herzke & Vorkamp, 2010; Domingo, 2012) or pollutants in the Baltic Sea region, where regular monitoring efforts have been in place since the 1980s. There remains a knowledge gap in the spatial trends and variability of contaminant loads at a global scale.

The goal of this review is to address four questions about global pollution: (1) do pollutant concentrations in marine food fish vary between geographic regions, and are these patterns consistent across pollutant groups? (2) does habitat use affect pollutant concentrations? (3) do pollutant levels increase from low to high trophic levels? and (4) have mean pollutant levels changed through time?

First, we elucidate patterns of pollutant concentration at large spatial and temporal scales. Previous studies have suggested that seafood captured in industrialized regions would have higher concentrations, both in large-scale (thousands of km) (Jensen et al., 1969; Burnett, 1971; Brown et al., 1998; Strandberg et al., 1998; Ueno et al., 2005) and small-scale (hundreds of km) (Albaiges et al., 1987; Adams & McMichael, 2007) analyses. However, predicting spatial patterning at global scales is difficult due to the physical properties of PBTs. Most PBTs are human-made chemicals that are semi-volatile, able to be absorbed on the water surface, and become subject to varied global currents and wind patterns. Despite potential global mixing, we may expect regional differences, similar to smaller scale studies. No study has attempted to resolve this question of spatial patterning across ocean basins.

Second, differences in habitat use among species might affect PBT accumulation, as the environment in which a fish spends the majority of its time can affect exposure potential. PBTs are hydrophobic and strongly adhere to sediments, presumably becoming more available to benthic than to demersal or pelagic species. A few studies have determined that feeding habitat plays a determinant role in the intake and subsequent accumulation of PBTs (Storelli, Giacominelli Stuffler & Marcotrigiano, 1998; Storelli & Marcotrigiano, 2000; Borga et al., 2004). Consequently, we expect benthic fish to have higher pollutant concentrations relative to demersal and pelagic taxa due to their increased direct exposure to PBTs.

Third, we test if a central process in ecotoxicology, biomagnification, is maintained when considering aggregated trends in PBT concentrations of numerous fish species at a global scale. Biomagnification is observed commonly, particularly among species in coastal ecosystems close to anthropogenic influence (Bayen et al., 2005). While terrestrial taxa show strong patterns of biomagnification (Kelly et al., 2007), patterns among marine taxa, especially marine fishes, are less clear. Many factors contribute to this heterogeneous landscape, confounded by food web complexity and length (Mizukawa et al., 2009) and the specific species examined (Fisk et al., 2001b). However, when considered across a broad suite of species and geographies, we may expect to see a signal of increasing pollutant levels with increasing trophic level.

Lastly, we assess the effectiveness of global mitigation programs to determine if mean pollutant levels have changed through time. Several national and international agencies are tasked with reducing and eliminating PBTs in the environment, including the United Nation’s Stockholm Convention. Although the Convention has the goal of assessing global trends of POPs, at this time only coarse regional analyses are available. Based on global mitigation efforts by major PBT-producing nations, we may expect to observe declines in mean concentration levels, particularly for legacy compounds (mercury and PCBs), over the past 50 years. To be clear, we will refer to mercury and PCBs as legacy compounds, as their production and use is more long-lived and effects better known than modern PBTs, which include organochlorine pesticides and flame retardants (PBDEs). Previous studies have reported successful reductions of PBTs in the environment, particularly in the Arctic and Baltic regions over longer time scales, at least a 10-year period (Andersson et al., 1988; Szlinder-Richert et al., 2008; Polak-Juszczak, 2009; Szlinder-Richert et al., 2009; Riget et al., 2010). At shorter time scales, these temporal trends are harder to distinguish statistically, but suggest decreases (Ikonomou et al., 2011). To the best of our knowledge, no global analysis of temporal trends among marine finfish in mean PBT concentrations has been conducted previously.

Here we provide a synthesis of available published data on muscle tissue of wild-caught, marine fish across the globe for the following contaminants: polychlorinated biphenyls (PCB), polybrominated diphenyl ethers (PBDE), two organochlorine pesticides, dichlorodiphenyltrichloroethane (DDT) and chlordanes (CHL), and mercury (Hg, MeHg). The objectives of this study are to synthesize global concentration levels and distribution of PBTs, to investigate the extent of regional, trophic, or habitat-related behavior patterning in data, and evaluate effectiveness of PBT mitigation efforts.

Methods

Literature review & data criteria

We collated data reporting concentrations of PBTs found in marine fish from across the globe. Two databases were used to identify available literature, Web of Science and Google Scholar, for the time period 1900–2013. Additionally, reference lists of selected papers were used to gather additional sources. Search results were constrained to peer-reviewed journals, grey literature, and government reports. Since we are interested in the effects of fish consumption on human health, and to standardize results, only studies reporting results from muscle tissue analysis were included. Studies were included only if raw data or mean concentrations of organochlorine pesticides (particularly CHLs and DDTs), polychlorinated biphenyls (PCBs), polybrominated diphenyl ethers (PBDEs), or mercury (total or MeHg) for individual species of marine fish were reported. The following search terms were used to browse the databases: fish* & (pesticide* OR organochlorine*) & marine; fish* & (polychlorinated biphenyl* OR PCB*) & marine; fish* & (polybrominated diphenyl ether* OR PBDE*) & marine; fish* & *mercury & marine. Finally, to be eligible for inclusion, sources had to contain both taxonomic identity, at least to genus level, and capture location; thus, market surveys reporting PBT concentrations of samples from unspecified capture locations were excluded. The results from the primary searches (n > 2,500) were reviewed, from which 303 papers contained requisite data allowing inclusion in this synthesis (Table S7).

Data collection

Data extracted from the resulting database included pollutant concentration and collection metadata, including capture date, capture location, tissue type, sample size, pollutant congeners, and taxonomic information. We define an individual data record as a mean concentration value calculated per pollutant group, per species, per year, per location for each study (Supplemental Information 1). For some pollutant groups, the specific congeners reported were not consistent across studies; therefore all congeners per pollutant group were summed to account for the total known exposure potential in an individual fish, or group of fish. The International Council for the Exploration of the Sea (ICES) has selected 7 PCB congeners (CB-28, CB-52, CB-101, CB-118, CB-138, CB-153 and CB-180), dubbed the ‘ICES 7,’ to be used in monitoring of PCBs in foods. These 7 congeners were selected due to their relatively uncomplicated identification and quantification in gas chromatograms and usual contribution to a substantial proportion of the total PCB content in environmental samples (Boalt et al., 2013).

When data were reported graphically, graph digitizing software (Plot Digitizer 2.6.3) was used to extract mean values of pollutants. Additionally, if the sampling date was not reported, collection was estimated to occur 2 years prior to publication date of the paper (sensu Hites, 2004).

All data were converted to ng/g wet weight for analyses, using the EPA Exposure Factors Handbook as a guide (Environmental Protection Agency, 2011). In many instances, concentration values standardized to the fish’s lipid concentration were reported. For lipid to wet weight conversions, concentrations were multiplied by species-specific lipid percentages to produce wet weight values. If no lipid percentage was reported, a literature search was conducted to find an estimated value of lipid concentration for each species. For dry weight to wet weight conversions, concentration values were multiplied by species-specific percent moisture values. If moisture values were not indicated, an 80% moisture value was used as an estimate for all fish species (Murray & Burt, 2001). Finally, for contaminant concentration values reported as “below detection limit,” a value reflecting 50% of detection limit was used (Environmental Protection Agency, 2000). If no detection limit as reported, a zero-value was used.

In some cases, mean contaminant concentration values were not included in the raw data of publications, only min (a), max (b), median (m) and sample size (n). From these given values, regardless of distribution, mean concentration (x¯) can be approximated as follows: x¯=a+2m+b∕4+a−2m+b∕4n (Hozo, Djulbegovic & Hozo, 2005).

The majority of the reviewed studies did not include morphometric data (body mass or length) for screened specimens. In the cases where collection method was reported, the specimens collected were representative of the associated fishery. Importantly, previous studies have reported correlations between length or body mass of fish and concentration value (Braune, 1987; Burreau et al., 2006; Hammerschmidt & Fitzgerald, 2006; Ikemoto et al., 2008). However, for the purpose of this review, body size was not taken into account. As such, this review captures pollutant loads based on available protein source from fish that are typically consumed by humans.

Figure 1 Distribution of data records.

A data record is defined as a mean pollutant concentration value for a distinct combination of species, sampling year, and sampling location. Size of pie charts reflects number of data records included in analysis for each region, with the color codes defining the class of PBT. The 14 global regions designated in study are NE Pacific, NW Pacific, SE Pacific, SW Pacific, Caribbean Sea, Gulf of Mexico, N Atlantic, S Atlantic, Indian, Mediterranean Sea, Baltic Sea, Red Sea, Arctic, and Southern.

Global regions

Global regions were defined as oceanic basins, accounting for major ocean currents and global wind patterns. Additionally, in order to prevent data overlap between defined global regions, anecdotal fisheries information was also used to delineate regions. Such regional designations offer reasonable boundaries for most species, though there are notable exceptions, e.g., bluefin tuna (Thunnus thynnus), which are capable of trans-oceanic migrations. Fourteen global regions thus were defined (Fig. 1); however, due to a paucity of data three regions were excluded from the analysis—the Arctic, the Southern Oceans, and the Red Sea. In order to create adequate sample sizes for statistical analyses, other regions were combined as follows; NE and SE Pacific combined to form the East Pacific region, NW and SW Pacific formed the West Pacific region, and the North Atlantic, South Atlantic, Gulf of Mexico, Baltic Sea, and Caribbean Sea formed the Atlantic Ocean region. The resulting groups defined for analysis included: East Pacific Ocean (EPO), West Pacific Ocean (WPO), Atlantic Ocean (AO), Mediterranean Sea (MS), and Indian Ocean (IO), which included >90% of the data from the original dataset. Regional summaries of PBT concentrations are provided in Table 1.

Table 1 Summary of regional pollutant means.

Pollutanta	CHL	DDT	Hg	PBDE	PCB	
Years	(1990–2011)	(1990–2012)	(1990–2010)	(1993–2012)	(1990–2012)	
Regionb	Mean [95% CI]	N	Mean [95% CI]	N	Mean [95% CI]	N	Mean [95% CI]	N	Mean [95% CI]	N	
EPO	4.0 [2.7, 5.6]	22	23.1 [17.1, 29.6]	27	372.7 [282.6, 492.4]	146	3.7 [2.2, 5.4]	53	44.9 [27.4, 65.9]	42	
WPO	6.8 [4.3, 9.7]	140	42.3 [18.0, 72.7]	145	524.7 [304.1, 810.0]	54	3.1 [0.9, 6.2]	75	37.0 [22.0, 54.5]	218	
AO	25.9 [11.7, 44.2]	24	9.8 [6.2, 14.3]	37	430.7 [380.8, 485.1]	482	1.7 [1.2, 2.3]	54	249.6 [138.8, 375.3]	108	
IO	2.3 [0.1, 8.2]	14	3.7 [0.9, 8.4]	27	434.2 [273.6, 688.2]	37	–	–	9.3 [1.8, 19.5]	31	
MS	1.8 [0.9, 2.9]	12	29.3 [19.5, 40.2]	45	418.5 [307.7, 556.3]	76	1.6 [0.4, 3.1]	20	26.5 [18.4, 36.5]	89	
Trophic level c	
H	7.1 [1.9, 17.9]	13	28.5 [7.2, 107.2]	15	462.9 [299.6, 680.5]	37	2.8 [0.6, 9.8]	14	89.1 [20.3, 231.0]	41	
P	7.6 [3.2, 13.8]	42	30.5 [11.8, 66.4]	54	492.6 [346.4, 679.6]	65	2.5 [1.0, 5.2]	36	83.3 [33.5, 159.5]	99	
MP	8.7 [5.2, 12.9]	130	31.8 [16.1, 55.5]	164	409.2 [355.6, 469.8]	417	2.7 [1.6, 4.1]	133	79.1 [44.4, 122.8]	265	
TP	7.6 [2.7, 16.2]	27	29.1 [11.7, 68.7]	47	429.7 [361.5, 509.4]	276	3.3 [0.9, 8.8]	19	85.1 [29.3, 166.5]	83	
Habitat type	
Benthic	8.1 [4.1, 12.1]	61	31.4 [13.2, 57.5]	75	416.1 [332.8, 489.6]	186	2.8 [1.2, 4.6]	48	83.1 [38.8, 130.7]	142	
Demersal	8.2 [13.6, 3.0]	121	31.0 [67.9, 13.0]	147	419.0 [515.1, 365.9]	365	2.9 [5.5, 1.0]	122	80.9 [155.3, 32.5]	238	
Pelagic	8.3 [5.1, 16.9]	30	28.0 [15.2, 59.6]	58	438.5 [358.1, 517.5]	245	2.5 [1.6, 5.6]	32	79.9 [45.7, 147.3]	108	
Notes.

a All data in ng/g, ww.

b Regions: EPO, East Pacific Ocean; WPO, West Pacific Ocean; AO, Atlantic Ocean; IO, Indian Ocean; MS, Mediterranean Sea.

c Trophic level: H, Herbivores; P, Primary consumer; MP, Middle consumer; TP, Top consumer.

Species classification

To address the effects of habitat and trophic variability on pollutant concentrations, habitat and trophic guild were determined using biological and ecological information obtained from FishBase (www.fishbase.org), an online aggregate database. Species habitat type was estimated from biological and ecological descriptions, defining species as pelagic, demersal, or benthic. Species were classified into four trophic guilds, based on trophic level, diet and food items reported in FishBase: herbivore (H), primary predator (P), middle predator (MP), and top predator (TP). In some cases, only genus was reported, in which case the same method was applied to generate a genus level trophic or habitat classification. All species names (common and scientific), habitat, and trophic designations can be found in Table S7.

Human health standards

Although toxicological health standards have been established by many organizations, for the purposes of this synthesis, only US Environmental Protection Agency (EPA) screening values for target analytes will be evaluated. Following EPA recommendations, the more conservative of the calculated values (noncarcinogenic) will be used because it is more protective of the consumer population. In order to assess human exposure and risk we used consumption standards developed by the EPA for both recreational fishers and subsistence fishers. These advisory guidelines assume a person to be 70 kg and have a 70-year lifespan, consuming an average of 17.5 and 142.4 g of fish per day for recreational and subsistence fishers, respectively (Environmental Protection Agency, 2000). EPA defines a screening value as a concentration threshold for target analytes in fish for which exceedance of these should signal more intensive monitoring or evaluation of human health risk.

Statistical analyses

Data from years 1990–2012 were used for statistical analyses, except for the evaluation of temporal decline of pollutant concentrations where all data were used (1969–2012). The shorter interval better represents modern exposure risk by avoiding potential inflation of concentrations due to variable usage and reporting patterns through time. We tested for differences across all species to see the effect of region, habitat type, and trophic level on pollutant levels. Change in pollutant levels over time was tested using linear regression. For statistical comparisons of groups, data transformation was not sufficient to apply parametric techniques, primarily due to the unbalanced nature of the design and the lack of homoscedasticity.

A non-parametric bootstrap approach, analogous to an analysis of variance, was used to evaluate differences between groups (Efron & Tibshirani, 1994). To create a null distribution (similar to F-distribution), samples of pollutant concentrations were drawn randomly with replacement from the total pool of concentration values, constraining group-specific sample sizes to the actual group size in the dataset. The mean squared error, i.e., the sum of the squared differences between group means and the global mean, was bootstrapped 10,000 times to create a distribution of mean squared errors. The test statistic, defined as the actual mean squared error of the data, was compared to the bootstrapped null distribution to determine whether the distributions were different among groups (including, but not limited to, the conclusion that the means were different among groups). Graphs are presented with group-specific means and 95% confidence intervals estimated from group-specific bootstrapping, with replacement. All analyses were performed using the statistical program R version 3.0 (http://www.r-project.org).

Results

Distribution of study sites

We collected 2,662 mean concentration values spanning years 1969-2012. Figure 1 details the total number of observations and the relative proportion of pollutants in each global region. There are hotspots of relatively high data density, particularly in regions where monitoring has been established, including the Baltic, northwest Pacific (China), and northeast Pacific (USA & Canada). A majority of the studies are from the northern hemisphere and Arctic, with relatively few studies available in the southern hemisphere and less developed nations. Within our dataset, the Atlantic and West Pacific Ocean regions make up 58% of the total data.

Despite the wide range of species (842 species) included in this review, sample sizes were largely homogeneous across trophic designations, habitat type and regions through decades (Figs. S1 and S2). Detailed information on sample sizes and data distributions are presented in Table S1.

Regional variability

We analyzed the regional variability of mean concentrations across five global regions. The Atlantic Ocean and West Pacific Ocean, compared to other global regions, generally had higher tissue concentrations for all pollutant groups. In contrast, the Mediterranean Sea and the Indian Ocean generally ranked lowest in mean concentration across all five pollutant groups (Fig. 2). Considering hundreds of species and a multitude of geographic locations, the spread between regional mean concentrations values rarely exceeded an order of magnitude within any one pollutant group. PCBs showed the highest spread among regional means (9.3–249.6 ng/g), whereas mercury has the smallest amount of spread among means (372.7–524.7 ng/g). Minimal differences between regions were found, with the exception of CHL (p < 0.001; n = 212) and PCBs (p < 0.001; n = 488), which have significant differences between regions (Table S2).

Figure 2 Regional variability of reported PBT concentrations.

(A)–(E) show concentration means and 95% confidence intervals around the mean, per global region (EPO-East Pacific Ocean; WPO-West Pacific Ocean; AO-Atlantic Ocean; IO-Indian Ocean; MS-Mediterranean Sea). Asterisks denote statistically significant differences between regions. Solid line: EPA recreational screening value (logPPB); Dashed line: EPA subsistence screening value (logPPB). Data include records from years 1990–2012.

To investigate underlying drivers of spatial differences, we compared regional mean levels within each trophic guild (Fig. 3). Within pollutant groups, we see inconsistent patterns of differences among regional means between trophic levels. For example, within PCBs and CHLs, the Atlantic Ocean has the highest levels of any region across all trophic groups. In spite of this, we failed to document consistent patterns of regional accumulation, even at this smaller scale. Statistically, no meaningful differences exist between regional means (Table S4), with the exception of mercury in herbivorous fish (p < 0.01; n = 37). Although some patterns can be observed, the highly variable means confirm the difficult nature in predicting trends at large spatial scales.

Figure 3 Regional variability of reported PBT concentrations within each trophic level.

Data presentation and labeling follow that of Fig. 2. Data include records from years 1990–2012.

In the regional analysis, pollutant levels were compared to EPA health standards (Fig. 2) for both recreational and subsistence fishers. For PBTs with established standards (Hg, DDTs, and CHLs), levels were generally near or above EPA recommendations. For all regions, mercury levels were well above the subsistence recommendation level (49 ng/g) and comparable to the recreational level (400 ng/g). Similarly, all regional mean concentration values for PCBs were above the recommended advisory level for subsistence fishers (9.83 ng/g), with only the Atlantic also exceeding the level for recreational fishers (80 ng/g). Contrary to mercury and PCBs, all regions fell below the subsistence fisher health standard (245 ppb) for DDT, and well below the recreational fisher level (2,000 ng/g). In the case of CHLs, all mean regional levels fell well below the screening values for both recreational (2,000 ng/g) and subsistence fishers (245 ng/g) by roughly an order of magnitude. For the remainder of pollutant groups (PBDE and CHL), EPA screening values have yet to be determined.

Habitat variability

We analyzed habitat type as a predictor of pollutant level, based on data from all capture locations (Figs. 4F–4J). For nearly all pollutants, differences in fish tissue levels between habitat types were not significant (Table S3). Mercury was the only pollutant to have significant differences between mean concentration as a function of habitat type (p < 0.001; n = 795). However, mercury behaved contrary to expectations; pelagic habitats (x¯=438.5, 95% CI [358.1–517.5]) were most contaminated and benthic zones (x¯=416.1, 95% CI [332.8–489.6]) the least. The difference between habitat means within each pollutant group was extremely low, never reaching an order of magnitude difference.

Figure 4 PBT concentrations as a function of trophic level and habitat of samples species.

(A)–(E) show concentration means and 95% confidence intervals around the means per trophic level (H-herbivores; P-low predators; MP-middle predators; TP-top predators). (F)–(J) show means and 95% confidence intervals per habitat. Asterisks denote a statistically significant differences between groups. Data includes records from years 1990–2012.

Trophic variability

We tested for biomagnification effects by comparing mean concentrations across trophic levels for all data combined (Figs. 4A–4E). We found significant differences in mercury concentrations between trophic levels (p < 0.001; n = 795), with a clear step-wise increase from low to high trophic levels. For the remaining four pollutants, no differences were found between trophic levels (Table S3). The inability to detect a signal was likely due to the high variability around the means as the 95% confidence intervals span up to two orders of magnitude for any given trophic level mean within a pollutant.

Temporal trends

Linear regressions were conducted to evaluate the effectiveness of global mitigation efforts to reduce pollutant levels. Temporal trends show reductions in all PBTs from 1969 to 2010 (Fig. 5). Despite high variability, all linear regressions report a significant negative slope in pollutant concentration over time (Table S5). To be sure the reported decline is not a result of high leverage points (e.g., a small number of anomalously high values in the early sampling years), the regression was tested using a shorter, most recent time frame, 1990–2012. These subsampled data revealed significantly negative trends with similar slopes for all PBTs, with the exception of DDT (Table S6). In this shorter time window, there was no evidence of systematic change in DDT concentrations across samples.

Figure 5 Temporal variability of pollutant concentrations from 1969–2012, inclusive of all trophic guilds and species.

Linear regressions reveal significant, negative trends of log-transformed concentrations through time for each class of PBT. Pollutant concentrations are presented as ng of pollutant per g wet weight of muscle tissue.

We subsequently evaluated the regional temporal change of each PBT group (Fig. S3). Specifically, we investigated whether each region showed a similar pattern of pollutant decline as shown in the global analysis. Regionally, there is a high amount of variability in slope values across all PBTs; no one region has a strong signal of increasing or decreasing pollutant levels. Only a few regressions showed a statistically significant decline (see Fig. S3). There does not seem to be any apparent trend as to which regressions significantly declined, likely due to the limited power and small sample sizes. CHL only reported a decline in the WPO; DDT only in the MS; Hg only in AO; PCB had more significant declines within EPO, WPO, AO, and MS. No significant declines were seen in any regions for PBDEs. However, there are interesting trends within particular PBT groups, specifically in CHL and PBDE. We see that CHLs have a pattern of slightly increasing levels of pollutants in the Atlantic and the Mediterranean Sea over the past 20 years. Similarly with PBDEs, half the regions with available data also report increasing levels.

The number of studies available from each decade can serve as an approximate timeline for the history of global PBT mitigation, particularly the initial awareness of negative health effects of each pollutant. However, a curious lack of data in the 1980s can be seen in all PBTs assessed in this review (Fig. 5 and Figs. S1–S2). For both PBDEs and PCBs, beginning in the early 2000s, there is a sharp increase in the volume of data, concurrent with the establishment of the UN’s Stockholm Convention and bolstered management and containment efforts across the globe. Although there is a decline in mean concentration value for all pollutants, variability at various time points remains quite high, with R2 values never exceeding 0.17. The variability of the mean concentration spans up to three orders of magnitude at any one time point for each PBT.

Discussion

The species included in this analysis encompass a range of characteristics (e.g., distribution, habitat, body size, trophic level, longevity) known to influence pollutant accumulation patterns. Despite such variability, our results show a variety of PBT-specific global differences in concentration across regions, trophic levels, and habitat type.

Regional variability

Based on previous studies, we expected that regions with more active production or use of PBTs would be linked with higher concentrations in finfish. It is well known that PBTs are capable of long-range transport, but it is unclear how this may affect regional differences at a global scale. As evidenced by many small-scale studies, pollution gradients exist, particularly around point source contamination (Litz et al., 2007; Gewurtz et al., 2011; Greenfield & Allen, 2013). Furthermore, previous literature suggests that a de facto divide exists between the Northern (NH) and Southern Hemispheres (SH), since the NH encompasses a majority of the industrialized nations and larger human populations, resulting in higher production and emissions of PBTs. With atmospheric exchange between the NH and SH being rather slow (∼1 yr), we expect minimal transport of PBTs between the two hemispheres.

Our analysis showed regional differences within some PBTs, but not all. For CHLs and PCBs, we see significant differences across space and consistent trends in regional accumulation. Similar regional patterns can be seen in CHLs, DDTs, and PCBs; for example, samples from the Indian Ocean revealed the lowest mean concentration levels within each class. Given the limited data available for PBDEs, likely due to their recent production, we begin to see regional variation but lack the statistical power to confirm. Finally, within mercury although a regional signal is not detected in the trophically aggregated data (Fig. 2), within disaggregated data (Fig. 3) we see regional variation within the herbivore guild. Although an analysis across hemispheres (north vs south) could provide further insight to regional pollutant trends, limited data from the southern hemisphere prevent such a detailed analysis.

This study notes the importance of scale in spatial toxicological investigations. The fact that variability exists between regions is not surprising, yet the ability to detect these differences decreases as scale increases. Particularly, as industrial activities are increasingly moving to Asia, elevated PBT emissions are shifting towards China and India (Lohmann et al., 2007). This shift in global emissions and production of PBTs can reduce regional signal, assuming that historical patterns were biased toward North America and Europe. Alternatively, pollutants may mix across water masses, thereby homogenizing environmental concentrations across regions and challenging the accuracy of spatial patterning predictions at a global scale. Regardless of mechanisms (similar introduction across the globe or rapid mixing of heterogeneous input), our data demonstrate that there are minimal regional differences in mean pollutant concentrations in marine finfish.

The lack of consistent monitoring of marine finfish across habitats and geographies, particularly studies targeting edible muscle tissue, is notable. There is an apparent trend to study estuarine and freshwater species, with few studies centered on pelagic species. This is likely due to the interest in understanding the mechanism behind PBTs bioaccumulation and trophic transfer. Further, due to the lipophilic nature of PBTs, much of the toxicological data available for marine fish is derived from liver tissue, rather than edible muscle tissue. Without a global standardized method of data collection and processing, evaluating the success or failure of mitigation becomes difficult as well as detecting regional differences.

Ecological and biological characteristics

In order to provide fish consumption recommendations, many studies have attempted to link species characteristics to pollutant accumulation patterns. The available literature suggests an expectation to see changes in pollutant concentrations based on trophic position and habitat type. Additionally, numerous studies report strong positive correlations of PBT concentrations and trophic position, particularly when paired with stable isotope analyses (Gobas et al., 1999; Fisk, Hobson & Norstrom, 2001a; Hoekstra et al., 2003). In terms of habitat type, the literature is less clear. A study from the South Adriatic Sea attributed differences in mercury concentrations to feeding habitat, with benthic species showing double the concentration relative to pelagic species (Storelli, Giacominelli Stuffler & Marcotrigiano, 1998). However, among the benthic species reported, the range of concentrations was quite high, with concentrations from some benthic species falling below that of pelagics. Our analysis revealed no conclusive evidence that mean pollutant levels were linked systematically to species’ trophic level or habitat. Mercury levels revealed the strongest patterns across trophic and habitat types. It is possible that the high level of variability around the means of the other PBTs clouds possible signals of trophic or habitat effects. In short, trophic level or habitat type alone is not a strong enough predictor to guide fish consumption recommendations at the global level for most classes of PBTs.

Pollutant level decline

To date, many published reviews and studies report that concentrations of many classes of PBTs have been decreasing in particular regions for particular species. Such declines are likely due to effective regulation by governmental and environmental protection agencies over the past few decades. However, these reviews are limited in scope, either temporally, spatially, or taxonomically. This synthesis provides a snapshot view of a worldwide decline of five classes of PBTs in marine finfish. The declines are significant, with a mean trend ranging across PBT classes from 100.2–100.5 (or 15–30%) decline per decade (Fig. 5). However, there exist large amounts of variation around these trends with the linear models across time describing <10% of the variance within each pollutant class. Even within recent years (2008–2012), we found significant variation in PBT concentrations. Reported concentrations of mercury and PCB span nearly four orders of magnitude when considering only “modern” studies. CHL, DDT and PBDE have lower variability around the mean compared to mercury and PCBs, but reported concentration still span approximately two orders of magnitude. When looking closer at the regional temporal change of PBT levels, clear trends are harder to distinguish due to reduced statistical power. Of the 24 possible regional analyses, only 7 regions across 4 PBT groups (CHL, DDT, Hg, PCB) reported a statistically significant change through time. PCBs contributed most to the significant declines, as opposed to PBDEs where no region reported a significant decline. This is likely a result of the more recent history of PBDE production and use and limited data prior to 1990. This high amount of variability can be contributed to a number of factors, including secondary sources of PBTs and environmental processing (Riget et al., 2010). Furthermore, inconsistent sampling techniques and locations can conceal possible trends, even within a region or pollutant. As a consumer, although the mean levels of PBTs tend to be lower now than in 1970, the high variance around this trend suggests that the chance of exposure via a contaminated fish remains non-trivial.

Conclusion

To the best of our knowledge, this is the first study to compile data on wild-caught marine fish with the goal of investigating spatial, life historical, and temporal patterns of tissue concentrations of PBTs at a global scale. Although we expected to see some level of regional distinction of PBTs, we found that regional trends are less distinct and confounded by finer scale details, including trophic level and habitat type. We conclude that the vagaries of global contamination cloud the predictability of toxin accumulation in marine finfish. The lack of standardized monitoring approaches, coupled with the globalization of seafood imports and exports, makes estimating the likely exposure to individual consumers based on market choices challenging. However, this analysis reveals the widespread and pervasive nature of persistent, bioaccumulative and toxic chemicals in seafood and the need to tackle these challenges.

In terms of human health, standards are developed in a singular fashion, evaluating risks for only one pollutant at a time. In reality, fish often contain multiple classes of PBTs simultaneously. Understanding additive effects of multiple exposures to PBTs is the next step in determining the “real” exposure risk to consumers, in all kinds of food. The quantification of these potentially synergistic (or antagonistic) effects could dramatically alter the health standards currently used. Furthermore, conclusive health standards still need to be determined for many of the PBTs, including many emerging containments not referenced in this review.

Based on the results of this synthesis, global efforts to reduce and eliminate PBTs appear to have been fairly successful, given the observed decline in the mean concentration values for PBTs over the past 50 years. However, the variability around this mean remains high, suggesting that the chance for exposure to a “contaminated” fish remains. Continued monitoring, stringent enforcement, and updated policies will be key to ultimately reducing the exposure to humans (and wildlife) to these human-made toxins. Additionally, novel policy and research must continue to address the potential negative effects of emerging contaminants entering our terrestrial and marine environments.

Supplemental Information

Supplemental Information 1 Supplemental Files

Click here for additional data file.

The authors would like to thank several anonymous reviewers for constructive criticisms of the manuscript. We would also like to thank S Nicklisch, C Edwards and Y Eynaud for comments on earlier drafts of the manuscript.

Additional Information and Declarations

Competing Interests

Author Contributions

Data Availability

The authors declare there are no competing interests.

Lindsay T. Bonito performed the experiments, analyzed the data, contributed reagents/materials/analysis tools, wrote the paper, prepared figures and/or tables, reviewed drafts of the paper.

Amro Hamdoun conceived and designed the experiments, reviewed drafts of the paper.

Stuart A. Sandin conceived and designed the experiments, analyzed the data, prepared figures and/or tables, reviewed drafts of the paper.

The following information was supplied regarding data availability:

The raw data was extracted from the literature, and all of the sources are cited in the text.

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
