# Peer review of "Evaluation of the global impacts of mitigation on persistent, bioaccumulative and toxic pollutants in marine fish"

_PeerJ, doi:10.7717/peerj.1573_

## Round 0.1 · original submission · Major Revisions

· Academic Editor

Major Revisions

This manuscript describes a detailed analysis of a large amount of environmental pollution data which provides information for the scientific and regulatory community. The data collection and statistical analyses are as thorough as can be expected given the experimental conditions. The results should be discussed more critically. For example, the regression analyses of environmental levels over time (Fig 5) appear to be highly influenced by one may be two data point for their statistical significance. Also, despite statistical significance, given the shallow slopes and huge variability, how much have levels actually gone down between 1960 and now? Have levels in certain areas gone up (e.g. arctic - it is also surprising that the investigators could not find arctic contamination data given the region has an active monitoring program-fish, seals, polar bears). Also, why was linear regression used. What if levels initially increased and only recently started coming down? A U shaped trend is entirely possible given the cloudy nature of the data.
Editorial comments:

Please redefine, replace or remove vague terms or jargon such as 'patterning', 'legacy', etc...as their meaning will not be apparent to all readers. Sentences are often convoluted, lack precision or are too vague.
e.g.: 'Mercury levels revealed the strongest patterns across trophic and habitat types.' or ' ..variability in pollutants..' or 'mean values of pollutants' What do these phrases mean?

Tidy-up expressions such as: concentration levels (=concentrations or levels).

Propositions should be corrected: at (time-scale) = on (or sometimes 'over'); through (time) = over. Propositions and other words are occasionally missing. Manuscript needs thorough editorial review.

Reviewer 1 ·

Basic reporting

In this paper a large dataset consisting of 2662 measurements of persistent organic pollutants (POPs) and mercury in marine fish from 303 papers were assembled. The data were then evaluated statistically to assess broad spatial and temporal trends as well as effects of habitat use and trophic level. These goals were quite ambitious and somewhat unrealistic given that data had to be aggregated from many species and geographic areas (eg the Atlantic Ocean is one region), there being almost no global studies of a single marine fish species. The only nearly global study that this reviewer has been able to find are the papers by Ueno et al on POPs in skipjack tuna (one is cited).

Experimental design

Despite this limitation the study largely succeeds in addressing these goals although the wide ranges of concentrations within and across regions limit the ability to demonstrate the importance of factors such as trophic level and habitat. The assembling of such a large dataset is unique and potentially useful if only to draw attention to the limited ability to assess global trends of pollutants which have undergone (eg PCBs, chlordane) or are undergoing (eg mercury) global bans or emission reductions. In this regard, the study, perhaps inadvertently, also demonstrates that studies focussed on spatial and temporal trends in the same species yield much more statistically powerful trends. Meta analyses of such studies (eg Riget et al (2010; 2011) for POPs and mercury) which included several datasets for marine fish provide more definitive results. Also Domingo & Bocio’sreview PCDD/Fs and PCBs in marine fish used a meta-data and assessment type of approach (Environ Int 2007). However both reviews were somewhat limited geographically given the preponderance of contaminants data from Europe, North America and the Arctic. The decision not to include contaminants trends in Baltic Sea was surprising considering that very strong datasets are available eg for herring (reported in HELCOM contaminant assessments). The authors also fail to note that assessment of global trends of POPs is a goal of the Stockholm Convention’s global monitoring program for effectiveness evaluation (http://chm.pops.int/Implementation/GlobalMonitoringPlan/Overview/tabid/83/Default.aspx) although admittedly it is not targeted at marine fish.

Validity of the findings

The weakest aspects of the paper are the sweeping statements about the knowledge of emissions, sources and biogeochemistry of the contaminants. The authors appear to be unfamiliar with some of the literature on global distribution and modelling of POPs and mercury. For example on line 47 it is stated that “estimates of global emissions and secondary sources of PBTs are poor or non-existent”. This is not the case for PCBs (see Brevik et al Environ Poll 2004; Sci Total Environ 2002) or for mercury (eg Mason et al (Environ Res 2012), Pacyna et al Atmos Environ 2010). Li et al (Environ Monit Assess 2005; other papers) have also estimated global inventories and emissions of chlorinated pesticides. The suggestion (line 76) that “ no study has attempted to resolve the question of spatial patterning across ocean basins” ignores the studies that have examined the oceanic distribution of POPs eg. Iwata et al ES&T 1993; Lohmann et al ES&T 2001; JGR 2006) and many global modelling studies which have included ocean waters (eg Lammel & Stemmler Atmos Chem Phys 2012).

Additional comments

Line 40. These references that support the statement that a “significant proportion of the world’s population is exposed to PBTs” are very USA centric. Yet there are many more globally oriented papers (eg for mercury see Chen et al EHP 2008; Environ Res 2012) and Domingo &Bocio (Environ Int 2007) for PCDD/Fs and PCBs.
Line 73. “rapidly volatilize” is not a correct term for describing the behavior of POPs such as most PCBs, DDT and chlordane compounds. Normally they are described as semi-volatile with the ability to be sorbed on surface and to revolatilize.
Line 82-83. Awkward sentence, seems to be combining POPs and mercury – meaning is unclear.
Line 91. The authors need to be more specific here re “patterns among marine taxa” because there are many examples of biomagnification of POPs and mercury in marine food webs but perhaps far less if homeotherms are left out of the calculation eg. Borga et al ET&C 2004.
Line 147. The “ICES 7” congeners are not the ones “commonly reported in toxicological studies” but rather are the ones used for monitoring PCBs in foods (for example). The co-planar PCBs would be examples of ones reported in toxicological studies.
Line 266. Shouldn’t results be limited to 2 or 3 significant figures given the known uncertainty or imprecision of measurement, especially of POPs.
Line 285. Why not use ng/g everywhere rather than “ppb” to be consistent with concentration data?
Line 436-438. The sentence implies that the Stockholm Convention has been successful because of the observed decline “over the past 50 years”. The SC only came into force in 2004 so cannot be credited with this decline, which is obviously due to national regulatory actions starting in the 1970s. Also it should be made clear that this is for POPs. Similarly global declines in mercury are attributable to national or regional actions.
Line 490. Missing information for this reference.

---

## Round 0.2 · accepted · Accept

· Academic Editor

Accept

Dear Authors,

Thank you for resubmitting your manuscript with major revisions, which were satisfactory for the reviewers. We congratulate you on providing the scientific community with pertinent and comprehensive data on global marine pollution with persistent organic pollutants.

Reviewer 1 ·

Basic reporting

no comments

Experimental design

no comments

Validity of the findings

no comments

Additional comments

The revised manuscript reads well and represents a significant improvement over the earlier version. The additional discussion on temporal trends (tracked changes version p 15) brings into focus some of the challenges in trying to do temporal trends with such as broad set of data. It's clear from Supp Figure 3 that declining trends for PCBs and DDT are mainly driven by the East Pacific and North Atlantic. So reporting global trends results less emphasisis on the important point that the trends are not global. The authors should consider using Supp Figure 3 in place of the current Fig 5. Whether the simple linear regression approach can be applied to this heterogeneous dataset is another issue. The usual approach for temporal trends of POPs eg see Stockholm Convention global monitoring guidance document, is to adjust the data for known covariates such as lipid. In the case of fish length or age can also be used although lipid normalization is often the most significant variable. Here the authors have converted all data to wet weight presumably because they didn't have lipid for all fish? I couldn't find any discussion of this although lipid to wet conversion is discussed.